# MAQL: Speeding up Q-learning with a model-assist

## Abstract

In reinforcement learning, model free methods such as Q-learning and policy gradient are extremely popular due to their simplicity but require a huge amount of data for training. Model based methods on the other hand, are proven to be sample efficient in various environments but are unfortunately computationally expensive. It is therefore only prudent to investigate and design algorithms that have best of features from both these classes of algorithms. In this work, we propose MAQL, a model-assisted Q-learning algorithm that is not only computationally inexpensive but also offers low sample complexity. We illustrate its superior performance to vanilla Q-learning in various RL tasks and particularly demonstrate its utility in learning the Gittins/Whittles index in Rested/Restless Bandits respectively. We aim to spur discussion on how model-assists can help boost the performance of existing RL algorithms.

## 1 Introduction

The main goal of Reinforcement Learning (RL) is to solve sequential decision making problem under uncertainty of the environment. There are broadly two approaches to solve the problem. In a model-free approach one interacts with the environment over possibly multiple episodes and aims to reach an optimal solution without building any model. In a model-based approach, one interacts with the environment to build a probabilistic model that best represents the real environment's dynamics and then solves the resulting Markov decision process (MDP) using dynamic programming methods such as value Iteration or policy iteration.

Model-based RL algorithms are useful in scenarios where data collection is expensive or time-consuming. One such popular algorithm is Posterior Sampling for Reinforcement learning (PSRL) Osband et al. (2013). PSRL adapts a Bayesian approach to model uncertainty and maintains a distribution over possible transition models. It then uses these models to plan and take actions, which allows it to effectively balance exploration and exploitation by considering a range of potential models. A recent work Sasso et al. (2023) showcases a deep implementation of PSRL (PSDRL) with comparable performances to state-of-the-art methods on the Atari benchmark. As one might expect, a key drawback of such model based methods is the huge computational cost involved in solving every interim model during the planning phase.

Model-free methods on the other hand are simple to implement and are computationally inexpensive, making them a popular choice among practitioners. Q-learning is one of the most popular value iteration based model-free RL algorithm and has been a topic of extensive study.Watkins & Dayan (1992) . Theoretical guarantees for a variant of Q learning with UCB exploration in an episodic setting have been established recently in Jin et al. (2018). Other popular model free methods include the policy gradient method Sutton & Barto (2018) where one optimizes a policy based metric (such as value function for a policy) by searching over the policy space using gradient ascent.

Empirical results Deisenroth & Rasmussen (2011) suggest model-free methods may require a large number of interactions with the environment to have a reasonably good performance. A tight sample complexity analysis of Q-learning Li et al. (2023) unveils its horizon dependence to be $\frac{1}{(1-\gamma)^4}$, which quantifies the negative impact of the widely studied overestimation bias (van Hasselt (2011)).

A natural idea to improve the sample efficiency of model-free RL algorithms is to include a model in the learning phase. This idea can be traced back to the Dyna algorithm Sutton (1991) which pro-

poses an integrated architecture for learning, planning and reacting. Dyna-Q builds a model from its interaction with the environment. It generating samples from the model and then learns for free from these additional samples, thereby improving upon the Q-learning algorithm. In the tabular setting, empirical estimates of transition dynamics and rewards function corresponds to the maximum likelihood estimation Luo et al. (2022) as these are unbiased estimates of the true transition dynamics and reward function.

Subsequently, there have been many variants of Dyna-Q such as Dyna-Q+ (with exploration bonus), Dyna-H (a heuristic planning algorithm based on Dyna) Santos et al. (2011) that have been proposed. Fitted Q-iteration Ernst et al. (2005) yields an approximation of the Q-function by interacting with the environment. Further works goes into finding a better function approximator for the Q function.Riedmiller (2005); Castelletti et al. (2012); Tosatto et al. (2017) Empirical Dynamic programming Haskell et al. (2013) introduces a class of algorithms using empirical estimation of Bellman operator instead of stochastic approximation and proves convergence of these new random operators to probabilistic fixed points. Kalathil et al. (2021) develops Empirical Q-Value iteration which estimates Q function using empirical estimates of the Bellman operator and showcases faster rate of convergence compared to stochastic approximation based algorithms. These methods assume access to a simulator to query any state-action pairs which is an idealistic scenario.

To address these above mentioned problems, we introduce a framework combining the best of both model-free and model-based approaches, MAQL model-assisted Q-learning. It involves using transition estimates to boost Q-learning updates for all visited state action pairs. This allows maximum utilization of the model's knowledge, instead of generating new samples. Our main contributions are summarized as follows:

- We introduce MAQL, model-assisted Q-learning algorithm which combines model-free and model-based approaches to improve Q-learning, along with its deep counterpart model-assisted DQN.

- We showcase empirical performance of MAQL over Q-learning in RL environments like grid-worlds , benchmark gym environment $Taxi$ and show how its DQN counterpart outperforms DQN.

- Adaptive usage of MAQL in Q-learning based algorithms to improve learning Gittins/Whittles index in Rested/Restless bandits settings respectively.

The remainder of the paper is organized as follows , Section 2 introduces the problem setting and Q-learning. In Section 3 we introduce Model-assisted Q-learning while Section 4 showcases Model-assisted DQN along with their respective experiments. Section 5 elaborates the usage of MAQL in a bandit setting.

## 2 PRELIMINARIES

### 2.1 MARKOV DECISION PROCESSES

A Markov Decision Process (MDP) is modelled by $(S, A, \gamma, P, R)$ where $S$ represents the set of states, $A$ represents the set of actions, $\gamma \in (0, 1)$ is the discount factor, $P : S \times A \times S \to [0, 1]$ is the transition kernel and $R : S \times A \to R$ is the reward function. We restrict our discussion to MDPs with only finite actions and states. On taking action $a \in A$ from state $s \in S$, we reach a new state $s' \in S$ with probability $P(s, a, s')$ and receive reward $R(s, a)$. Value function $V^\pi(s)$ denotes the expected total discounted reward from policy $\pi$ starting from state $s$. The corresponding state action value function for the state action pair $(s, a)$ is denoted by $Q^\pi(s, a)$. For finite MDPs, there is always atleast one policy that is better than or equal to all other policies. These optimal values are solutions to the Bellman optimality equation Sutton & Barto (2018).

$$Q^*(s, a) = \max_\pi Q^\pi(s, a)$$
$$= \mathbb{E}\left[R_{t+1} + \gamma \max_{a'} Q^*(S_{t+1}, a') \mid S_t = s, A_t = a\right]$$
$$= R(s, a) + \gamma \sum_{s'} P(s, a, s') \max_{a'} Q^*(s', a')$$

## 2.2 Q-LEARNING

Q-learning Watkins (1989)is one of the most famous RL algorithms applied for various control tasks owing to its flexible nature and simplicity. Q function tells the expected return one gets starting from state $s$, taking action $a$ and following policy $\pi$. It can be seen as an asynchronous implementation of the Robbins-Monro procedure for finding fixed points Robbins & Monro (1951). To implement this in a online fashion for $(s_t, a_t, r_t, s_{t+1})$, we apply the update as follows:

$$Q_{t+1}(s_t, a_t) = Q_t(s_t, a_t) + \alpha_t(s_t, a_t)(r_t + \gamma \max_{a' \in A} Q(s_{t+1}, a') - Q_t(s_t, a_t)) \tag{1}$$

If the sequence of the algorithm $(s_t, a_t, r_t, s_{t+1})$ visits each state, action infinitely and if the learning rate is an adaptive sequence satisfying the Robbins-Monro condition $\sum_{t=0}^{\infty} \alpha_t(s, a) = \infty$ and $\sum_{t=0}^{\infty} \alpha_t^2(s, a) < \infty$ , then with probability 1 , $Q_t(s, a) \rightarrow Q^*(s, a)$.

## 3 MODEL-ASSISTED Q-LEARNING

We now lay the framework for our model-assisted Q-learning (MAQL) algorithm. We build the model using a simple empirical estimator for the transition matrix and the reward function, henceforth denoted by $\hat{p}_t(s, a, s')$ and $\hat{r}_t(s, a)$ at a time step $t$ for all $(s, a, s') \in S \times A \times S$.

$$\hat{p}_t(s, a, s') = \frac{N_t(s, a, s')}{N_t(s, a)} \qquad \hat{r}_t(s, a) = \frac{\sum_{i=0}^{t} r_i \mathbb{1}[(s_i, a_i = s, a)]}{N_t(s, a)} \tag{2}$$

where $N_t$ counts the visits (state action pairs $(s, a)$ and $(s, a, s')$) until time t and $\mathbb{1}(E)$ returns one if $E$ is true otherwise zero. Now we describe the Q-iteration step similar to value iteration using the above defined estimators,

$$Q_{t+1}(s, a) = \hat{r}_t(s, a) + \gamma \sum_{s' \in S} \hat{p}_t(s, a, s') \max_{a' \in A} Q_t(s', a') \tag{3}$$

The idea of the algorithm is simple and can be seen as alternating between traditional Q-learning,

---

**Algorithm 1** Model-assisted Q-learning

**Input**: Exploration policy $\epsilon_t$ , Learning rate $\alpha_t$
**Parameter**: A set of episodes $C$ , Step $k \in N$
**for** $t_{max}$ episodes **do**
    $a_t \leftarrow$ Select action based on $\epsilon$-greedy policy
    Get new state $s_{t+1}$ and reward $r_t$ from state $s_t$
    Update $\hat{p}_t(s, a, s'), \hat{r}_t(s, a)$ following Equation 2.
    Update $Q(s_t, a_t)$ according to Equation 1
    **if** $t \in C$ **then**
        Perform k steps of Q iteration on $Q(.)$ acc. to Equation 3 for all visited $(s, a) \in S \times A$
    **end if**
**end for**

---

which updates the Q values asynchronously from the samples and value iteration like Q-updates happening synchronously over all visited state-action pairs. $C$ contains a set of episodes when the model-based updates should be executed. In Dyna-Q style algorithms, the built model is used to generate additional samples to perform Q learning on the same (learning for free). It can be a condition as once in every 40 episodes or a schedule to switch completely to value iteration style updates. Here, the updates in true Q learning and sampled Q learning bring the same order of change to the Q values, whereas in model-assist : as all visited $(s, a)$ pairs are updated k times, it will result in a better iterative scheme to reach the optimal function. $C$ and $k$ needs to be appropriately set as per the model's ability to learn the environment dynamics. Such a model-based update would allow for a more sample-efficient way to reach the optimal $Q^*$.

As a preliminary example, we consider a GridWorld environment of $|S| = 24$ and $|A| = 4$ where the goal state is selected to be one of the states. We assume that obstacles have high negative rewards(-100), the goal state has a high positive reward(+100), while other states have low negative

rewards(-1). The rewards and transition dynamics is not known to the agent. The transitions are probabilistic (there is a 0.1 chance the agent doesn't reach the desired next state). We begin by comparing the $Q(s, a)$ values for MAQL and Q learning in Fig.1. We also compare the relative error in Q values for Q-learning and MAQL in Fig.1 .We use hyper-parameters $k = 10$ , $c = 2, 10$ , $\gamma = 0.9$. Jin et al. (2018) shows the advantage of using $\alpha_t = \frac{H+1}{H+n(s,a)}$ in sample efficient

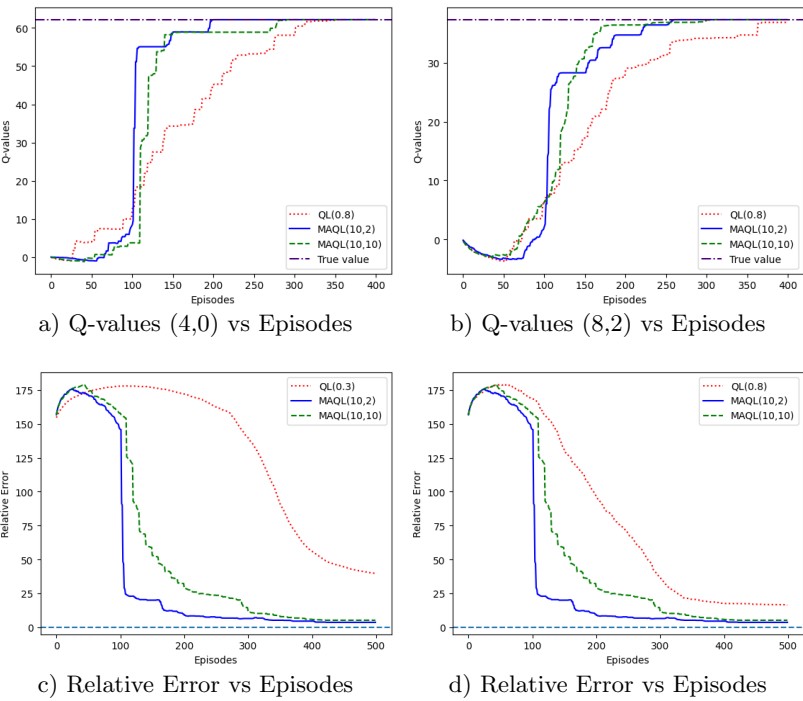

a) Q-values $(4,0)$ vs Episodes      b) Q-values $(8,2)$ vs Episodes

c) Relative Error vs Episodes      d) Relative Error vs Episodes

Figure 1: a) shows Q-values for state-action pair (3,0) vs Episodes while b) and c) shows Relative Error $|Q_k - Q^*|$ vs Episodes for different learning rates in a GridWorld environment, the shaded region represents the confidence interval whereas the solid line represents the mean over 20 runs

Q-learning (here $H$ is taken as 500 ), where $n(s, a)$ denotes the number of times the state action pair $(s, a)$ is visited. With adaptive learning rate, we see that $c = 2$ is better than $c = 10$ as more updates occur It is clear enough that $\left(\frac{k}{c}|S||A|\right)$ updates occur in every episode , once the model is accurate enough, increasing $\frac{k}{c}$ would give better results. Model-assisted Q-learning clearly outperforms Q-learning in all the cases. It is also visible that Q-learning 's convergence rate varies greatly on different learning rates whereas model-assisted Q-learning shows stable outperformance.

We also demonstrate the performance of our algorithm on the Gymnasium environment *Taxi-v3*. This is a deterministic environment with $|S| = 500$ and $|A| = 6$. Each episode starts randomly at one of the 300 possible states. A successful drop results in a reward of $+20$ , wrong pickup-drop results in $(-10)$ while $(-1)$ is the return for every step taken. An episode terminates after the drop or by using truncated lengths (maximum length of episode is set as $200$). Model-assisted Q-learning is tuned with $c = 25$ and $k = 1$ with the adaptive learning rate. We train it using $\epsilon$-greedy policy for 500 episodes and then test on the environment by using the greedy policy. On testing, Q-learning with a slower learning rate attains a success rate of $(56.51 \pm 3.76\%)$, Q-learning with a faster learning rate attains $(84.88 \pm 3.02\%)$, Q-learning with the adaptive attains $(76.92 \pm 2.18\%)$ whereas Model-assisted Q-learning manages to attain **100%** just after the same amount of training episodes. This clearly states, our algorithm is able to learn the environment perfectly within the given episodes. All results are averaged over 20 runs.

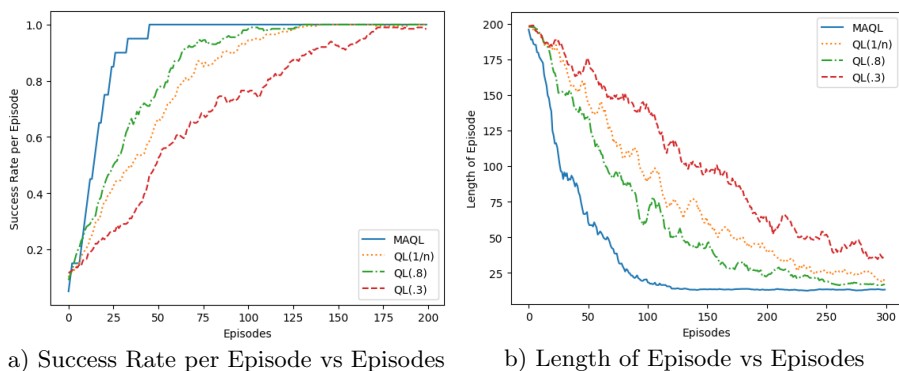

a) Success Rate per Episode vs Episodes        b) Length of Episode vs Episodes

Figure 2: Performance on Benchmark environment $Taxi$

## 4  MODEL-ASSISTED DQN

DQN has enjoyed major success in learning policies for various control and RL tasks by approximating the Q function with a neural network Mnih et al. (2013). However, they often require extensive interaction with the environment and can suffer from instability and inefficiency due to the approximation errors in value estimation. We introduce model-assisted DQN which allows the network to learn from the traditional targets as in Q-learning and targets from the model-based estimates.

---
**Algorithm 2** Model-based update $MBUPDATE()$

---
Create $\mathcal{B}$ with targets $y$ by performing k steps acc. to Equation 3 for all visited $(s, a) \in S \times A$
**for** epochs = 1, $E$ **do**
    Calculate loss function $L$, $\left[(y_j - Q(s_j, a_j; \theta))^2\right] \forall y_j, s_j, a_j \in \mathcal{B}$
    Optimize $L$ wrt $\theta$ on all data in $\mathcal{B}$ using gradient descent
**end for**
$\theta^- \leftarrow \theta$
Return $\theta^-, \theta$

---

Using such model-based updates help in mitigating the overestimation bias present in traditional Q-learning by using refined Bellman targets, $\hat{r}_t(s, a) + \gamma \sum_{s' \in S} \hat{p}_t(s, a, s') \max_{a' \in A} Q_t(s', a')$. The model of the environment is learnt using empirical estimates here. Other approaches to build the model like Gaussian Processes Deisenroth & Rasmussen (2011) are quite common in literature. The function $MBUPDATE()$ can be implemented efficiently by using prioritized sweeping to decide which state-action pairs need to be updated instead of the whole $S \times A$ space.

In DQN style algorithms, the agent takes actions by a $\epsilon$ greedy policy from values approximated by Q-network. There are usually two networks, online and target to ensure stability in learning values. A soft update is performed between the two networks after every $C$ steps. A replay buffer $\mathcal{D}$ is maintained to randomly sample mini-batches to learn Q-values, which can be guided by a squared loss between the targets and current values. The weights of the network can be optimized by using gradient descent. Such learning occurs iteratively over time and the Q-network approximates the true Q-values. Now model-assisted DQN adds to DQN $SWITCH$, a set of episodes deciding when model-based updates are performed along with $k$ in the $MBUPDATE()$ function which decides how many steps of Q-iteration is to performed.

We consider a Windy GridWorld $|S| = 100$ and $|A| = 4$, where wind distracts the agent from taking the desired action with some probability. The training episodes is set to be 100, meaning data collection for both model-assisted DQN and DQN stops after 100 episodes. $SWITCH$ is done at episodes 25, 50 and 75. The Q-network is a simple 3 layer neural net, batch size is set as 10, target network is copied to online Q-network after every 10 steps, trained using the same $\epsilon_t$ greedy

**Algorithm 3** Model-assisted DQN

Initialize Q-network with random weights $\theta$, target network with weights $\theta^- \leftarrow \theta$
Initialize replay buffer $\mathcal{D}$ to capacity $N$, a set of episodes to switch updates $SWITCH$
**for** episode = 1, $M$ **do**
    Initialize state $s_0$
    **for** $t = 1, T$ **do**
        Select action $a_t$ using $\epsilon$-greedy policy:
        Take action $a_t$, observe reward $r_t$ and next state $s_{t+1}$
        Update estimators $\hat{p}_t(s, a, s')$, $\hat{r}_t(s, a)$ following Equation 2.
        Store transition tuple $(s_t, a_t, r_t, s_{t+1})$ in replay buffer $\mathcal{D}$
        Sample random mini-batch of transitions $(s_j, a_j, r_j, s_{j+1})$ from $\mathcal{D}$
        Compute target $y_j$ for each mini-batch sample:

$$y_j = \begin{cases} r_j & \text{if } s_{j+1} \text{ is terminal} \\ r_j + \gamma \max_{a'} Q(s_{j+1}, a'; \theta^-) & \text{otherwise} \end{cases}$$

        Perform a gradient descent step on loss $\left[ (y_j - Q(s_j, a_j; \theta))^2 \right]$
        Every $C$ steps, update target network: $\theta^- \leftarrow \tau\theta + (1 - \tau)\theta^-$
    **end for**
    **if** episode $\in SWITCH$ **then**
        Update $\theta^-, \theta$ using $MBUPDATE()$
    **end if**
**end for**

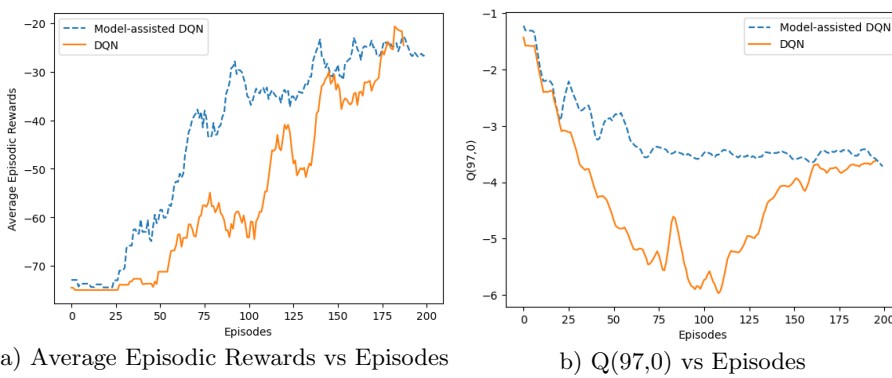

a) Average Episodic Rewards vs Episodes         b) Q(97,0) vs Episodes

Figure 3: Comparison of Model-assisted DQN and DQN on Windy GridWorld

policy. Figure 3 a) clearly shows that model-assisted DQN performs better than DQN in learning the optimal values faster while b) shows it helping the network mitigate the overestimation bias for a state-action pair.

## 5 APPLICATION TO MARKOVIAN BANDITS

An interesting case of MDPs are the Markovian Bandit setting where each arm can be modelled as a Markov chain governed by unknown state dynamics. They extend the classical multi-armed bandit problem, offering more complex scenarios to model real-world applications like single-machine scheduling, resource constraint problems. The setting where the state of a Markov chain stays frozen unless it's played is rested, while restless arms' state may continue to evolve regardless of the player's actions.Tekin & Liu (2012)

## 5.1 RESTED BANDITS

Gittins and Jones in 1974 proposed a Gittins index policy Gittins (1979) which is shown to be optimal in pulling the arms that would maximize the cumulative expected discounted rewards collected. Duff was the first to propose a Q-learning based algorithm to learn the Gittins index that uses a novel 'restart-in-state-i' interpretation for the problem, while the earlier work all assumed a known setting of the transition dynamics. Recent works like QWI, QWINN converges to the gittins index but seems to be noisy due to continuous updates for both the action values.

Dhankar et al. (2024) suggests a tabular QGI algorithm to compute the gittins index and has shown better performance than other formulations in scheduling applications. QGI relies on a two-timescale stochastic approximation for Q updates which eventually leads to the retirement value converging to the gittins index. It is governed by the below equations.

$$Q_{n+1}^x(s_n, 1) = Q_n^x(s_n, 1)(1 - \alpha(n) + \alpha(n)\Big(r(s_n) + \gamma \max\{Q_n^x(s_{n+1}, 1), M_n(x)\}\Big) \quad (4)$$

$$M_{n+1}(x) = M_n(x) + \beta(n)\left(Q_n^x(x, 1) - M_n(x)\right) \quad (5)$$

where $Q_n^x(s_n, 1)$ refers to the Q function's value at state $s_n$ at nth time step with respect to reference state $x$, $M_n(x)$ refers to the retirement value which is equal to $Q_n^x(s_n, 0)$ for all $S$ and $\alpha(n), \beta(n)$ are their respective learning rates. Let's introduce a tabular model-assisted QGI with hyper-parameters $para$ and k, i.e the model based Q-iteration step is done k times once in every para time steps and the Q iteration step is written with respect to state $x$,

$$Q_{n+1}^x(s, a) = \hat{R}(s, a) + \sum_{s' \in S} \hat{P}(s, a, s')\gamma \max\{Q_n^x(s', 1), M_n(x)\}) \quad (6)$$

We illustrate the performance of the algorithms on a slightly modified version of the restart problem from Robledo Relaño et al. (2024). Here passive arms do not undergo state transitions. $|S| = 3$ and there are 2 homogeneous arms. Reward function $r(s) = (0.9)^s$ and there is an unknown transition dynamics in case of active action :

$$P(s'|s, 1) = \begin{pmatrix} 0.3 & 0 & 0.7 \\ 0.3 & 0.3 & 0.4 \\ 0.3 & 0.3 & 0.4 \end{pmatrix}$$

The Gittins index can be analytically computed by any of the methods given in Chakravorty & Mahajan (2014), $G(s) = (0.9, 0.8343, 0.7911)$. We have run experiments for QGI with different learning rates. It seems to converge to the true Gittins value regardless of learning rate but rate of convergence greatly depends on it. Whereas model-assisted QGI is run with $para = 10$ , k = 1. Since this is a simple toy setting, performance is not greatly affected by $para$ and k. Model-assisted QGI seems to be robust not only in convergence also in rate of convergence to different learning rates.

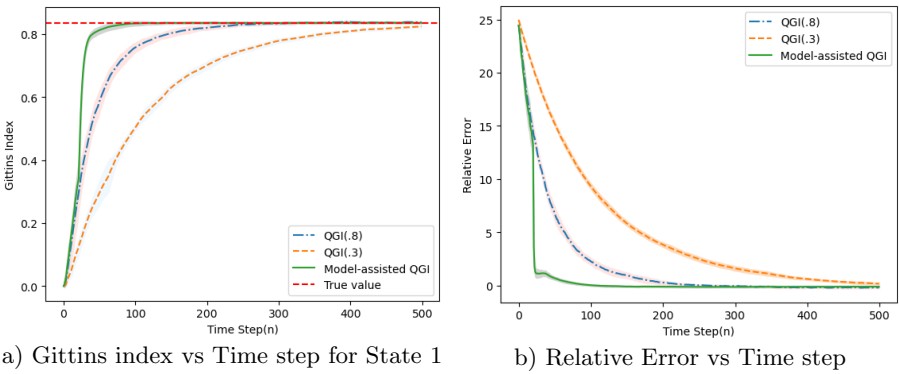

a) Gittins index vs Time step for State 1  b) Relative Error vs Time step

Figure 4: a) compares the convergence of Gittins Index for QGI and Model-assisted QGI, b) compares the Relative error $|Q_k - Q^*|$ for both the algos for the toy setting. All the plots are averaged for 20 runs and shaded region shows the confidence interval.

Let's consider a setting where the agent has to pull 1 out of 5 homogeneous arms and each arm is modelled by a Markov chain of 10 states with $R(s) = 0.1(1 + s)$ and arms are passive when not pulled, when they are pulled it is governed by the following dynamics,

$$
P(s'|s,1) = \begin{pmatrix}
0.2 & 0.1 & 0.3 & 0.1 & 0.3 & 0.0 & 0.0 & 0.0 & 0.0 & 0.0 \\
0.0 & 0.1 & 0.2 & 0.3 & 0.25 & 0.15 & 0.0 & 0.0 & 0.0 & 0.0 \\
0.0 & 0.0 & 0.15 & 0.15 & 0.25 & 0.15 & 0.3 & 0.0 & 0.0 & 0.0 \\
0.0 & 0.0 & 0.0 & 0.1 & 0.3 & 0.3 & 0.25 & 0.05 & 0.0 & 0.0 \\
0.0 & 0.0 & 0.0 & 0.0 & 0.1 & 0.3 & 0.3 & 0.25 & 0.05 & 0.0 \\
0.0 & 0.0 & 0.0 & 0.0 & 0.0 & 0.2 & 0.1 & 0.3 & 0.1 & 0.3 \\
0.3 & 0.0 & 0.0 & 0.0 & 0.0 & 0.0 & 0.2 & 0.3 & 0.1 & 0.1 \\
0.25 & 0.05 & 0.0 & 0.0 & 0.0 & 0.0 & 0.0 & 0.3 & 0.1 & 0.3 \\
0.15 & 0.25 & 0.3 & 0.0 & 0.0 & 0.0 & 0.0 & 0.0 & 0.15 & 0.15 \\
0.3 & 0.2 & 0.1 & 0.3 & 0.0 & 0.0 & 0.0 & 0.0 & 0.0 & 0.1
\end{pmatrix}
$$

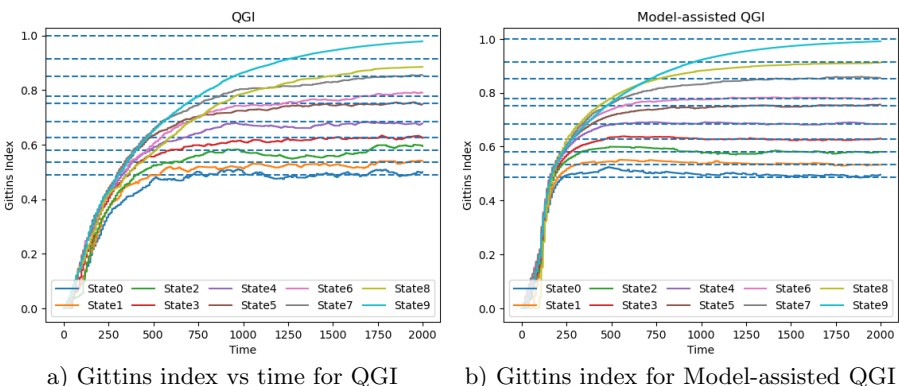

a) Gittins index vs time for QGI  b) Gittins index for Model-assisted QGI

Figure 5: Comparison between the convergence of Gittins Index for QGI and Model-assisted QGI

We select this setting as the true gittins values are closer for few states and one can observe in QGI, it can lead to sub-optimal decisions before the convergence, whereas in model-assisted QGI not only we are able to reduce the variance for the same learning rate but also the ordering of states is achieved sooner than QGI leading to better decisions in early episodes too.

## 5.2 RESTLESS BANDITS

RMABs(Restless Multi-armed-Bandits) first introduced by Whittle in 1988 has found various applications in healthcare, inventory routing and networking.Whittle (1988) It is similar to the rested setting but even in the case of passive action states undergo transition. The decision maker has to pull K out of N arms. A relaxed version of the setting stating that the arms are indexable then the optimal policy is Whittle index policy that relies on calculating Whittle index for each of the arms, and activating in every decision epoch the arms with the highest K Whittle indices. QWI introduced in Robledo Relaño et al. (2024) seems to enjoy better performance than other RL based algorithms like NeurWIN Nakhleh et al. (2021). It relies on a faster timescale to update the state action Q-values and a relatively slower timescale to update the Whittle indices, explained in the below equations.

$$
Q_{n+1}^x(s_n, a_n) = (1 - \alpha(n))Q_x^n(s_n, a_n) + \alpha(n)*
$$
$$
((1 - a_n)(r_0(s_n) + \lambda_n(x)) + a_n r_1(s_n) + \gamma \max_{v \in \{0,1\}} Q_x^n(s_{n+1}, v)) \quad (7)
$$

$$
\lambda_{n+1}(x) = \lambda_n(x) + \beta(n)(Q_n^x(x, 1) - Q_n^x(x, 0)) \quad (8)
$$

The Q iteration step for these are given by:

$$
Q_{n+1}^x(s, a) = (1 - a)(\hat{r_0}(s) + \lambda_n(x)) + a\hat{r_1}(s) + \gamma \sum_{s' \in S} \hat{P}(s, a, s') \max_{v \in \{0,1\}} Q_x^n(s', v) \quad (9)
$$

Now we use this Q iteration step and plug it in QWI algorithm as before in the rested case using a switching condition $C$.

We take the example of circular problem introduced in Fu et al. (2019) with a state space $|S| = 4$. In this problem, with an active action the process remains in its current state with probability 0.6, or increments positively with probability 0.4. Similarly, with a passive action the process remains in its current state with probability 0.6, or decrements negatively with probability 0.4.

The reward function does not depend on the action performed, but only on the state, being $R(0) = -1, R(1) = R(2) = 0, R(3) = 1$. Using a discount parameter of $\gamma = 0.9$, the theoretical values of Whittle index for each state are $\lambda(0) = -0.4390, \lambda(1) = 0.4390, \lambda(2) = 0.8652, \lambda(3) = -0.8652$.

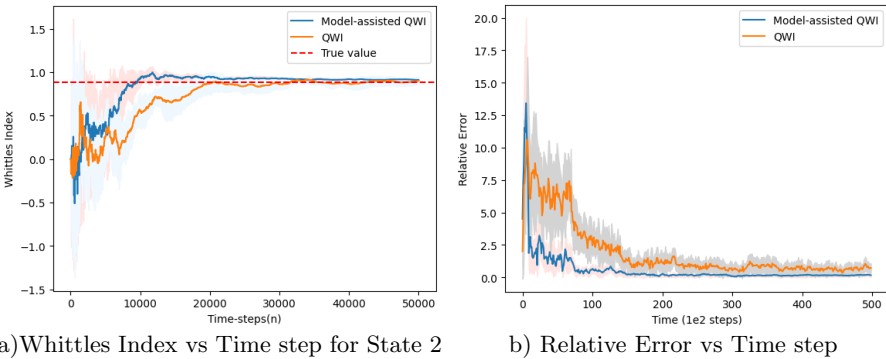

a) Whittles Index vs Time step for State 2     b) Relative Error vs Time step

Figure 6: a) compares the convergence of Whittles Index for QWI and model-assisted QWI, b) compares the Relative error for both the algos for the Circular problem where one has to pick 1 out of 3 arms. All the plots are averaged for 20 runs and shaded region shows the confidence interval.

The learning rate is taken as described by the authors of QWI for both the algorithms. We can see that model-assisted QWI performs better than QWI by converging to the optimal Q values faster and by reducing the number of sub-optimal episodes. The initial deviations might come due to model's incorrect estimates of state dynamics but later as more interaction with the environment occurs, estimates move closer to the true dynamics, thereby causing the Q values and Whittles Index also converge faster.

## 6   CONCLUSION

In this paper, we present Model-assisted Q-learning a framework combining model-free and model-based methods to improve the sample efficiency of Q-learning on finite MDPs. We also showcase similar success in model-assisted DQN over traditional DQN and its adaptive usage in learning Gittins/Whittles index in rested and restless bandits setting respectively. The theme of these algorithms is to improve the sample complexity of existing Q-learning methods as collecting data in certain environments can be time-consuming and expensive. As our methods model the environment using exact parameters empirically, it is restrictive in $|S|$ suffering from the curse of dimensionality. Using non-parametric approaches like Gaussian Processes for approximating environment or parametric function approximators like neural networks is a direction for future work. Using such model-assists for natural policy gradient style methods for policy evaluation instead of interacting with the environment sounds interesting.

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
