# OpenReview forum: "MAQL: Speeding up Q-learning with a model-assist"
_ICLR.cc/2025/Conference — ICLR 2025 Conference Withdrawn Submission_

### Official Review · Reviewer_ku9x · 2024-10-21

**Soundness:** 2
**Presentation:** 2
**Contribution:** 2
**Rating:** 3
**Confidence:** 4

**Summary:**

This paper seeks to enhance the performance of Q-learning by incorporating the state-counting defined in equation (2), and evaluates this approach through a series of simple experiments.

**Strengths:**

This paper is self-contained and well-structured.

**Weaknesses:**

1. **Methodological Limitations**: The proposed approach involves counting states, actions, and next states from experiences to improve the prediction of future Q-values based on rewards. However, this method faces a fundamental limitation when applied to environments with continuous state or action spaces, where the number of possible (state, action, next state) pairs becomes infinite, making such counting impractical. As a result, the applicability of this method is extremely restricted.

2. **Insufficient Experimental Validation**: The experimental results are not sufficiently convincing. While the paper uses DQN, the experiments are conducted only in very simple toy environments. To demonstrate the robustness of the method, I believe experiments on more complex benchmarks, such as Atari games, are necessary.

3. **Poor Writing Quality**: The writing in this paper is not good, with numerous typographical, citation, and mathematical expression errors. For example, the term "model-free" is inconsistently used—appearing as "model free" in the abstract, but "model-free" in lines 41-42. There is also an incorrect citation in lines 37-38, 46-47, 48-49, while an incorrect definition of the reward function exists in lines 96-97, and a typographical error with "atleast" in lines 101-102.

**Questions:**

How to adopt your algorithms into a environment with continuous action space and/or state space?

---

### Official Review · Reviewer_KSBb · 2024-11-01

**Soundness:** 2
**Presentation:** 2
**Contribution:** 2
**Rating:** 3
**Confidence:** 3

**Summary:**

The paper studies dqn with model-based approach. The transition probability is estimated by maximum likelihood method, i.e., how often it has occured.

**Strengths:**

1. The paper studies model-assisted DQN based on model-assisted Q-learning. The model-assisted Q-learning has been studied in quite detail in theoretical sense, which is not properly discussed in this paper. Please refer to weakness 1.

**Weaknesses:**

1. The model-assisted DQN approach requires $|\mathcal{S}|^2|\mathcal{A}|$ memory space to store the transition probability.  As the authors mentioned, it would be better to show that prioritized sweeping or Gaussian Process to show that this approach is still valid while not degrading the performance too much.

**Questions:**

1. I understand that this is an empirical paper but misses most of the literature on model-based Q-learning approach [1,2,3, 4] in the tabular setting. In particular, in [1], the update in equation (2) and (3) is exactly the one considered in [1]. Even though the details might slightly different in implementing the algorithm, the key update is the same.  Furthermore there are somewhat practical implementations based on such theory, e.g., [5], and can the authors provide comparison with it?


**References**

[1] Lim, Han-Dong, HyeAnn Lee, and Donghwan Lee. "Finite-Time Error Analysis of Online Model-Based Q-Learning with a Relaxed Sampling Model." arXiv preprint arXiv:2402.11877 (2024).


[2] Szita, István, and Csaba Szepesvári. "Model-based reinforcement learning with nearly tight exploration complexity bounds." Proceedings of the 27th International Conference on Machine Learning (ICML-10). 2010.

[3] Brafman, Ronen I., and Moshe Tennenholtz. "R-max-a general polynomial time algorithm for near-optimal reinforcement learning." Journal of Machine Learning Research 3.Oct (2002): 213-231.

[4] Tor Lattimore and Marcus Hutter. Near-optimal pac bounds for discounted mdps. Theoretical Computer Science, 558: 125–143, 2014.

[5] Henaff, Mikael. "Explicit explore-exploit algorithms in continuous state spaces." Advances in Neural Information Processing Systems 32 (2019).

---

### Official Review · Reviewer_nqGx · 2024-11-01

**Soundness:** 2
**Presentation:** 2
**Contribution:** 2
**Rating:** 5
**Confidence:** 3

**Summary:**

This paper introduces MAQL and Model-assisted DQN algorithms that learn by integrating a learned model into Q-learning and DQN, respectively. The authors experimentally demonstrate the performance improvements of MAQL over standard Q-learning in GridWorld and Gymnasium Taxi environments and of Model-assisted DQN over DQN in a Windy GridWorld. The authors also demonstrate an adaptation of MAQL to improve the learning of indices in rested and restless bandit settings.

**Strengths:**

The authors propose a simple but effective algorithm that is applicable to various environments including both structured and unstructured problems.

**Weaknesses:**

1-	Limited exploration of MAQL’s performance across diverse, complex environments. While the empirical results demonstrate the algorithm's improvement over standard Q-learning and DQN in relatively simple (toy) environments (like grid-worlds, the Taxi domain), the paper does not investigate its scalability in larger, high-dimensional state spaces, where model-based assistance could potentially face the curse of dimensionality. For example, could you evaluate performance on Atari games or continuous control tasks from the MuJoCo suite?

2-	The authors compared performance against simple baselines such as standard Q-learning and DQN and did not compare performance against highly relevant algorithms such as Dyna-Q. The authors could show the significance of their algorithm by comparing against other algorithms like Double-QL, Dyna-Q, Rainbow DQN, etc. This is especially important since their proposed algorithm is similar in spirit to Dyna-Q.
- In particular, how does MAQL compare to Dyna-Q in terms of sample efficiency?
- How does MAQL handle the exploration-exploitation trade-off compared to Double-QL?
- Also, could you compare the computational efficiency of MAQL to these baselines, given that one of MAQL's claimed advantages is being computationally inexpensive?

3- Lack of detailed ablation studies, e.g., investigating the sensitivity of MAQL to variations in model accuracy.
- For example, how sensitive is MAQL to specific hyperparameters like the frequency of model-based updates (parameter 'c' in Algorithm 1) or the number of Q-iteration steps (parameter 'k')?

**Questions:**

N/A

---

### Official Review · Reviewer_sf1a · 2024-11-04

**Soundness:** 3
**Presentation:** 2
**Contribution:** 2
**Rating:** 3
**Confidence:** 4

**Summary:**

This paper introduces MAQL, a novel algorithm that combines the strengths of model-based and model-free Reinforcement Learning (RL) to improve the sample efficiency of traditional Q-learning. MAQL alternates between traditional Q-learning updates and model-based value iteration-like updates. The authors also provide an extension of MAQL to DQN and show empirical results on some simple environments.

**Strengths:**

- This paper shows a way to combine model-free and model-based methods, which seems promising for RL.

**Weaknesses:**

- The empirical results are quite lacking. There are many model-based algorithms out there that should be compared to, as baselines, but the authors only compare to Q-learning. Not only that, there are also many variants of Q-learning that could be worthy baselines, but the authors also did not include those. The same is true for the set of DQN experiments. Because of this, I'm not adequately convinced in the usefulness of these methods. (For example, why not use some of the baselines mentioned in the intro?)

- In MA-DQN, the tabular equation (2) is used for the model learning. But in DQN settings, we are in a setting with large state (or action) spaces, so this method of model learning is not going to scale well. I wonder how the authors would address this major issue?

- Because the results are only shown for relatively simple environments, the above concern is not really tested. I'd recommend the authors test their method on real settings where DQN would be applied.

**Questions:**

- Can we say anything theoretically about this algorithm? It seems to me that under certain assumptions, one should be able to show convergence by leveraging standard arguments.

---

### Note · Authors · 2024-11-15

**Comment:**

We would like to express our sincere gratitude to the reviewers for their invaluable feedback and insights. After careful consideration, we have decided to withdraw our submission to further develop and refine our paper based on these constructive comments. Thank you once again for your time and thoughtful evaluation.

**Withdrawal Confirmation:**

I have read and agree with the venue's withdrawal policy on behalf of myself and my co-authors.